Sloth metabolism may make survival untenable under climate change scenarios

Cliffe Rebecca N. Rebeccacliffe06@gmail.com 1 2
Ewart Heather E. 1 3
Scantlebury David M. 4
Kennedy Sarah 1
Avey-Arroyo Judy 5
Mindich Daniel 5
Wilson Rory P. 2
1 The Sloth Conservation Foundation , Hayfield , Derbyshire , United Kingdom
2 Swansea Lab for Animal Movement, Biosciences, College of Science, Swansea University , Swansea , Wales , United Kingdom
3 School of Biological Sciences, The University of Manchester , Manchester , United Kingdom
4 School of Biological Sciences, Queen’s University Belfast , Belfast , Northern Ireland , United Kingdom
5 The Sloth Sanctuary of Costa Rica , Limon , Costa Rica
Kaburu Stefano
Electronic publication date: 2024 Sep 27
Publication date: 2024
Volume: 12
Electronic Location ID: e18168
Received 2024 Jan 23; Accepted 2024 Sep 3
Copyright: ©2024 Cliffe et al.
Copyright year: 2024
Copyright holder: Cliffe et al.
License: This is an open access article distributed under the terms of the Creative Commons Attribution License, which permits unrestricted use, distribution, reproduction and adaptation in any medium and for any purpose provided that it is properly attributed. For attribution, the original author(s), title, publication source (PeerJ) and either DOI or URL of the article must be cited.
License URL: https://creativecommons.org/licenses/by/4.0/

Keywords: Choloepus hoffmanni, Climate change, Metabolism, Energetics, Conservation, Resting metabolic rate, Sloth

Funding: Indiegogo Crowdfunding Campaign Sloth Conservation Foundation This research was funded by donations to an Indiegogo crowdfunding campaign and the Sloth Conservation Foundation. The funders had no role in study design, data collection and analysis, decision to publish, or preparation of the manuscript.

==============================
Although climate change is predicted to have a substantial effect on the energetic requirements of organisms, the longer-term implications are often unclear. Sloths are limited by the rate at which they can acquire energy and are unable to regulate core body temperature (Tb) to the extent seen in most mammals. Therefore, the metabolic impacts of climate change on sloths are expected to be profound. Here we use indirect calorimetry to measure the oxygen consumption (VO2) and Tb of highland and lowland two-fingered sloths (Choloepus hoffmanni) when exposed to a range of different ambient temperatures (Ta) (18 °C –34 °C), and additionally record changes in Tb and posture over several days in response to natural fluctuations in Ta. We use the resultant data to predict the impact of future climate change on the metabolic rate and Tb of the different sloth populations. The metabolic responses of sloths originating from the two sites differed at high Ta’s, with lowland sloths invoking metabolic depression as temperatures rose above their apparent ‘thermally-active zone’ (TAZ), whereas highland sloths showed increased RMR. Based on climate change estimates for the year 2100, we predict that high-altitude sloths are likely to experience a substantial increase in metabolic rate which, due to their intrinsic energy processing limitations and restricted geographical plasticity, may make their survival untenable in a warming climate.

Introduction

Climate change is having a profound effect on the metabolism and behaviour of organisms (Deutsch et al., 2015; Dillon, Wang & Huey, 2010; Laloë et al., 2014; Levy et al., 2017; Parmesan & Yohe, 2003) both directly (e.g., increased thermoregulatory demands Dillon, Wang & Huey, 2010; Oswald & Arnold, 2012) and indirectly (e.g., through changes in resource availability or trophic interactions Fuller et al., 2021). While these changes can sometimes have a positive effect on population viability (Loe et al., 2021; Laloë et al., 2014) the specific consequences of a warming climate on the survivability of many different species often remain uncertain. Although many animals have the capacity to compensate for a degree of temperature variation through genetic adaptation (Bradshaw & Holzapfel, 2001), physiological and behavioural plasticity (Boyles et al., 2011; Fuller et al., 2016), or modifications of distributions (Parmesan & Yohe, 2003; Tourinho et al., 2023), these options are biologically implausible for some (Colwell et al., 2008; Malcolm et al., 2006). There is thus a need for a better understanding of the survivability of species in response to increased temperatures (Parmesan, 2006) coupled with identification of vulnerable areas where conservation strategies may be necessary to prevent extinction (Laloë et al., 2014).

As ambient temperatures (Ta) change, the energetic demands on animals also change (McNab, 2002). The thermoneutral zone (TNZ) is the range of ambient temperatures within which a homeothermic animal does not need to expend extra energy to maintain its core body temperature (Tb). For the majority of homeotherms, this typically means that, as Ta’s rise above the TNZ, energetically costly mechanisms are instigated in order for Tb to remain stable (Lowell & Spiegelman, 2000; Nagy, 2005; Pat, Stone & Johnston, 2005; Schmidt-Nielsen, 1997). Ectotherms, however, experience an exponential increase in metabolic rate with Ta due to the increase in rates of biochemical and enzymatic reactions (Daniel et al., 2010; Levy et al., 2017; Schulte, 2015). This explains why climate change is considered to be invoking large metabolic costs on tropical-dwelling ectotherms, exacerbated by the already high temperatures in these regions (Dillon, Wang & Huey, 2010; Seebacher, White & Franklin, 2014). Thus, while considerable work is now examining the impacts of climate change at mid-high latitudes (e.g., Deutsch et al., 2015; Pauchard et al., 2016), species living in the tropics are likely to be much less resilient to fluctuations in temperature, in part due to their evolutionary histories in comparatively stable climatic environments (Christian, Bedford & Schultz, 1999; Dillon, Wang & Huey, 2010; Doucette et al., 2023; Pounds, Fogden & Campbell, 1999).

Sloths (Bradypus spp and Choloepus spp) are poikilothermic tropical mammals (Geiser, 2004; Irving, Scholander & Grinnell, 1942; McNab, 1978; Montgomery & Sunquist, 1978). They have an unusually low and variable Tb and utilise postural adjustments in order to exploit favourable microclimates within the canopy and thereby regulate Tb (Britton & Atkinson, 1938; Montgomery & Sunquist, 1978; Urbani & Bosque, 2007). This is considered to be a strategy to reduce the energetic requirements of thermoregulation in animals that subsist on an extremely low-calorie diet (Cliffe et al., 2015; Cliffe et al., 2018; Geiser, 2004; Nagy & Montgomery, 1980; Pauli et al., 2016). With little energy at their disposal, sloths are presumed to exist within a narrow and finely tuned energy budget, in which minimal expenditure is linked to minimal energy intake. This, combined with a limited dispersal ability (Peery & Pauli, 2012), means that the metabolic implications of even a small degree of climate change could have profound implications on the persistence of sloth populations (Tourinho et al., 2022; Tourinho et al., 2023).

Previous work has shown that lowland-dwelling sloths from the genus Bradypus are capable of invoking temporary metabolic depression in response to high temperatures (Cliffe et al., 2018). This physiological flexibility is likely to facilitate a reduction in both Tb and energy expenditure through an overall reduction in metabolic heat production. In comparison, sloths from the Choloepus genus have much broader phenotypic and geographical plasticity (Gilmore, Da Costa & Duarte, 2001; McNab, 1978; McNab, 1985; Montgomery & Sunquist, 1978; Pauli et al., 2016; Vendl et al., 2016), and inhabit both highland and lowland tropical forests. The metabolic response of these animals to changes in Ta, however, is unknown. Animals living at higher altitudes tend to have physiological and morphological adaptations to cope with a colder climate (Broekman et al., 2007; Pichon et al., 2013; Wasserman & Nash, 1979; Yu et al., 2016) and this is apparently the case in Choloepus sloths inhabiting highland forests as they have longer, thicker, and darker pelage than their lowland counterparts (Enders, 1940; McNab, 1985). We hypothesised that this increase in insulation would reduce the thermal conductance of high-altitude sloths and should, theoretically, result in them having a higher overall body temperature and, consequently, a higher metabolic rate than sloths from low-altitude regions. This, combined with their lack of geographical plasticity, may leave high-altitude populations in a vulnerable position when faced with a warming climate, especially given that atmospheric warming in highland forests is amplified relative to the lowlands (Pounds, Fogden & Campbell, 1999).

To test this theory, we investigated the change in resting metabolic rate (RMR) and Tb of Choloepus hoffmanni sloths originating from both highland and lowland rainforests when exposed to a range of different Ta’s (18 °C–34 °C). We additionally recorded changes in Tb and posture over several days in response to natural fluctuations in Ta. We then used the resultant data to predict the metabolic and Tb impact of future climate change on the different populations.

Materials & Methods

Ethics

This research was approved by the Swansea University Animal Welfare & Ethical Review Process Group (AWERP), and the Costa Rican government and associated departments (MINAE, SINAC, ACLAC) permit number: R − 033 − 2015

Resting metabolic rate (temperature manipulation in the metabolic chamber)

Sample and study site

Twelve adult C. hoffmanni sloths (eight male, four female) were chosen for metabolic measurements. All of these were captive animals that, although wild-born, were being maintained permanently at the Sloth Sanctuary of Costa Rica (N09°47′56.47″W 082°54′47.20″) after being rescued as they were unsuitable for release. This sample size was chosen as it encompassed all available sloths at the sanctuary that were deemed suitable for participation in the project (i.e., adult, healthy, not pregnant, had been maintained in captivity for >18 months and with accurate origin location records). Four of the sloths (three male, one female) originated from high-altitude locations while the remaining eight sloths originated from lowland areas (Table S1). All metabolic testing was completed during daylight hours in the Sloth Sanctuary veterinary clinic between May and September 2015. Ten of the sloths were sedated prior to metabolic testing in order to minimise stress and facilitate handling. Each individual was sedated using 1 mg/Kg of ketamine (Ketamina 50®, Holliday Scott) and 0.008 mg/kg of dexmedetomidine (Dexdomitor®, Zoetis) administered intramuscularly. Sedation was reversed before the sloth entered the metabolic chamber using 0.008 mg/kg of anti-sedante (atipamezol; Antisedan®, Zoetis). Two sloths were not sedated as a control (one male, one female).

Measurement of body temperature

A miniature temperature logging device (iButton®, Thermochron, Dallas Semiconductors; Maxim Integrated Products, Inc., Sunnyvale, CA, USA) (model DS1922L (±0.0625 °C)) was inserted into the rectum of nine of the sloths using a gloved digit and lubricant. The logger was calibrated prior to use by immersion into a temperature-controlled water bath and programmed to record temperature every 30 min (Cliffe et al., 2018). Sloths defecate only once a week, storing faeces in an anal pouch. Rectal insertion of the temperature logger was therefore deemed the least-invasive, non-surgical method of obtaining accurate core temperature values. If faecal pellets were found in the anal pouch of the animal, then these were removed prior to logger insertion to ensure the most accurate temperature readings.

Measurement of resting metabolic rate (RMR)

Metabolic data were collected as previously described in (Cliffe et al., 2018). Specifically, prior to measurements, all sloths were weighed (E-PRANCE® Portable Hanging Scale (±0.01g)). They were then placed in an 87-L Perspex® metabolic chamber (55 cm long × 45 cm high × 35 cm wide). The chamber was placed in a temperature-controlled water bath which was covered with a polystyrene lid. The water bath (95 cm × 85 cm × 75 cm), also made from Perspex®, was lined with black plastic sheeting and supported with an exterior metal frame. Within the metabolic chamber, there was a branch for the animal to hold on to, and from which it could comfortably suspend itself upside down. There was a small window in the plastic sheeting (a ‘peep’-hole) through which the sloth could be observed without it being disturbed by the observer.

Oxygen consumption (VO2) was measured using an open-flow system with an upstream flow meter. Fresh air from outside was pumped into the chamber (AIR CADET® Barnant, model 420-1902; Barnant, Barrington, IL, USA), via a copper coil submerged in the water bath, at rates of between 4 and 12 L/min. Flow rate was adjusted to the mass of the sloth to ensure that the depression in oxygen concentration within the chamber remained in the range 0.2–0.8% (Speakman, 2013). The flow was measured using a flow meter (ICEhte10 platon flow meter 1-12L/min; ICEoxford Limited, Oxford, UK) which was factory calibrated and checked prior to use using a mass-flow generator (Sable Systems Flowkit 100; Las Vegas, NV, USA). The incurrent air flow rate was measured before drying. The system was checked for leaks using a dilute solution of soapy water. The air inlet was located on the opposite side of the chamber to the air outlet to ensure an adequate mixing of air within the chamber. Air leaving the chamber was subsampled at 200 ml/min and then dried (using Drierite) before entering an oxygen and carbon dioxide analyser (FoxBox Field Gas Analysis System, Sable Systems International, Las Vegas, NV, USA). The length of tubing leading from the metabolism chamber to the gas analysers was 0.5 m. The lag time for the analyser reading to equilibrate when the tubing was placed into the chamber to subsample the gasses was less than 1 min. The analyser was factory calibrated and set to 20.95% oxygen before each animal was measured. Fresh air readings were recorded at the start and the end of each run to correct for analyser drift. Any drift in the analyser was assumed to be linear for baseline correction. An acclimatization period of ∼150 min was allowed at the beginning of each experiment for any sedation to wear off, for each sloth to become accustomed to the chamber, for  Tb to adjust to the chamber temperature and for the chamber gases to equilibrate (McClune et al., 2015). The animals were observed continuously through the peep hole (for welfare reasons and to make sure they weren’t showing any signs of stress). During measurement periods (i.e., following temperature adjustment periods and when gas concentrations had stabilised), oxygen and carbon dioxide concentrations were recorded manually at two-minute intervals. A total of 12 experimental runs were made (Table S2). An ‘experimental run’ refers to a series of measurements from one animal, taken during the course of a day.

VO2 (ml/min−1) was calculated as: (1) VO2=FR⋅FiO2−FeO2−FeO2⋅FeCO2−FiCO21−FeO2

where FR is the flow rate, FiO2 is the fractional amount of O2in the chamber incoming air, FeO2 is the fractional amount of O2 in the outgoing air, FiCO2 is the fractional amount of CO2 in the incoming air and FeCO2 is the fractional amount of CO2 in the outgoing air (Lighton, 2008). Values were corrected for standard temperature and pressure. Metabolic rates were calculated using a conversion factor of 20.1 joules per millilitre of oxygen, which is correct for an obligate herbivore such as the sloth (Schmidt-Nielsen, 1997).

Values for resting metabolic rate (RMR) were compared with allometrically predicted values for terrestrial mammals as cited in (Kleiber, 1961; White & Seymour, 2003).

Temperature manipulation

Temperatures within the chamber were manipulated following the protocol described by (Cliffe et al., 2018). This was achieved by varying the temperature of the water bath which contained two electric water heaters (Grant water bath heater circulator) and two water fans which stirred the water in a clockwise direction around the metabolic chamber. The temperature within the chamber was measured using a copper-constantan thermocouple and monitored on a Tecpel 307P Dual Input Digital Thermometer (0.1 °C). Chamber temperature was recorded at four-minute intervals throughout the duration of each experimental run. The first three experimental runs were undertaken with the chamber maintained at constant temperature. The remaining 9 experimental runs had the chamber temperature directly manipulated. Following the initial ∼150-minute acclimatization period, the temperature of the metabolic chamber was increased incrementally in 2-degree steps i.e.: 16−19 °C, 20−23 °C, 24−26 °C, 27−29 °C, 30−32 °C, and 33−35 °C by varying the temperature of the water bath. These temperature brackets were selected as they encompass the most extreme range of ambient temperatures to which Choloepus sloths are naturally exposed in the wild.

The length of time animals spent at each temperature increment was sufficient to allow both equilibrations of gases within the chamber, and for the animal Tb to adjust to the new Ta. Typically, animals spent 60 min adjusting to each 2-degree temperature increment. Following the c.60-min adjustment period, when sloths were seen to be at rest and the gas concentrations had stabilised, RMR readings took place and recordings were taken every 2 min for a further 10 min. RMR values were then calculated from the mean of these 5 values. In nearly every case, the sloths were inactive, apart from slow postural adjustments. As a control, the empty chamber was taken through 5 different temperature increments on three separate occasions prior to testing with animals. During these control tests, temperatures were recorded from twelve different locations within the chamber (Cliffe et al., 2018).

The effect of natural fluctuations in Ta on Tb and posture

Thirty-four C. Hoffmanni sloths (seventeen males, seventeen females, six high-altitude, twenty-eight low-altitude) had pre-calibrated iButton® temperature loggers inserted rectally. No sedation was necessary, and all logger insertions were carried out without removing any sloths from the enclosures. The loggers were programmed to record temperature every 30 min.

All sloths were housed in individual standardised enclosures measuring 5.3 m2 with a shelf (114 cm by 61 cm) and 13 horizontal climbing bars. Sloths were fed twice daily at 7am and 2pm. The enclosures were outdoors, exposing the animals to natural fluctuations in Ta although, to ensure uniform temperatures and minimise possible microclimate differences, all enclosures were covered by a metal roof to prevent access of rain or direct sunlight. Although levels of non-visible light such as ultraviolet (UV) were not monitored in this study, the metal roofing should have standardised and minimised these effects. Three further temperature loggers were uniformly distributed throughout the enclosures in order to measure Ta.

Following temperature logger insertion, visual surveys were completed on all sloths at 2-hour intervals for 48 h. Posture was graded on a scale of 1–6 (1 = tight ball, 6 = all limbs spread) (Cliffe et al., 2018; Muramatsu et al., 2022). Temperature loggers were collected opportunistically when the sloths defecated. The mean time that the temperature loggers were retained in the rectum was 3.1 days. Six temperature loggers were never retrieved and were presumed to have been washed away during cleaning of the enclosures. Consequently, data presented are from twenty-eight sloths (fifteen males, thirteen females, three of these being high-altitude sloths, twenty-five low-altitude sloths).

Statistical analysis

Resting metabolic rate (temperature manipulation in the metabolic chamber)

All statistical analyses were conducted in R (version 4.3.1) (R Development Core Team, 2016). The percentage difference between the measured and allometrically predicted values was calculated by dividing the difference by the allometric prediction. The relationship between RMR, ambient temperature and altitude was determined using a hierarchical linear mixed model (LMM) fitted using the ‘lmer’ function from the “lme4” package (Bates et al., 2015). The LMMs were first tested to confirm basic assumptions were met—normality of residuals and homoscedasticity were analyzed using residual diagnostic plots (i.e., normal Q–Q plot) (Fig. S1). Body temperature, body mass and sex were entered as covariates and animal ID as a random factor to allow for repeated measurements within individuals. Two LMMs were fitted—the first included ambient temperature as recorded by raw temperature measurements collected in the trials; the second applied a categorical representation of ambient temperature using high (≥32 °C) and low (<32 °C) values. The latter model was included as a separate model to ensure the correlation between the two representations of ambient temperature values did not skew a single model. The categorical representation of temperature was included to measure effects of and interactions between more meaningful temperature classes (i.e.,  ≥32 °C) and altitude on RMR, to test the hypothesis that altitude origins predict sloths’ metabolic responses to changes in temperatures. Two LMMs were fitted using maximum likelihood (ML) during model selection to account for the random effects—both models included all variables and data, the only difference being one included Ta as a continuous variable and one included Ta as a categorical variable. Each model was selected with stepwise backwards model selection, whereby one explanatory variable/interaction was tested at a time using ANOVA and those variables/interactions with p > 0.05 were removed until all variables/interactions in the final model were significant (p < 0.05). The final presented models were then refitted using restricted maximum likelihood (REML) (Table 1). For the analysis, we only used data from the nine trials in which sloths were exposed to a broad range of ambient temperatures (metabolic chamber periods >3 h) to determine the effect of ambient temperature on RMR. RMR was also compared between high- and low-altitude sloths (which had and had not been sedated prior to entering the metabolic chamber) across all ambient temperatures, as well as in the high and low temperature categories, using a two-sample t-test or a Mann–Whitney test. A Shapiro–Wilk test was first used to test the normality of the distribution of the data (normally distributed data were interpreted with a t-test, and non- normally distributed data with a Mann–Whitney test).

Table 1 Fixed effects of ambient temperature (continuous –LMM1; categorical [<32 °C and ≥32 °C] LMM2), body temperature, altitude origin, sex, and body mass on RMR. Ta is the only variable listed from LMM1; all other variables are from LMM2.

Dependent variable	SE	t-value	p-value	
Predictor variable				
LMM2 (RMR)		
Ta (cont., LMM1)	0.43	8.70	0.001	
Tb	3.31	5.22	0.001	
Altitude:Ta (cat.)	14.13	3.72	0.001	
Ta (cat.)	9.90	−1.50	0.14	
Altitude	26.14	−1.46	0.18	
Sex	27.49	−0.57	0.59	
Body mass	29.65	−0.38	0.71	
Body mass: Ta (cat.)	23.58	−1.70	0.10	

The effect of natural fluctuations in Ta on Tb and posture

Rectal Tb and natural Ta were recorded at 30-minute intervals. Due to the high likelihood of temporal autocorrelation in temperature data, a generalized additive mixed model (GAMM) was used to test the relationship between Ta and Tb. The GAMM included Julian day and time, as well as mass, sex, and altitude as covariates using the “gamm4” package (Wood & Fabian, 2022). The ‘acf’ function was used to test for autocorrelation in the temperature data; where autocorrelation was found, the GAMM model was refitted to correct for autocorrelation using the ‘corAR1’ function. The ‘mgcv’ package was used to test whether the relationships between Ta and Tb, Tb and time of day, were linear or polynomial and the best fit model was used to analyse relationships. The standard smooth function set in the models was a cubic regression spline with automatically set knots. Differences in Tb between high- and low-altitude sloths were examined using a Mann–Whitney test after conducting a Shapiro–Wilk test to assess the normality distribution of the data. The mean time lag between ambient and core body temperature was determined by calculating the average time between maximum/minimum ambient temperature and maximum/minimum core body temperature for each individual. The effect of Ta and Tb on posture was examined using a GAMM—two different models were tested given the correlation between Ta and Tb; the best fit model was identified as the one with the lowest AICc score.

Projected impacts of climate change

To estimate the projected impacts of climate change on the body temperature (Tb) of sloths, we employed a bootstrap method adjusted for sample size differences to account for the uncertainty around our predictions. The rate of change of Tb ( °C/min) for both high- and low-altitude sloths was plotted against the difference between Tb and Ta. We used the resultant regression equations to model the projected Tb increase for high- altitude and low-altitude sloths if the climate warmed by an average of 2 °C. We simulated 1,000 predictions for each projected Tb value by drawing from a normal distribution centred on the mean of the projection with a standard deviation equal to the standard error of the model predictions. We derived 95% confidence intervals for the predicted Tb values by taking the 2.5th and 97.5th percentiles of the bootstrap distributions (Efron & Tibshirani, 1994) (Fig. S2).

To estimate the projected impacts of climate change on the RMR of sloths, we used a similar bootstrap approach adjusted for sample size (Table S3). Individual regression equations were calculated for high- and low-altitude sloth RMR as a function of Ta (calculated for temperature brackets: 19−23 °C; 23−27 °C; 27−29 °C; 27−29 °C; 29−32 °C; 32−34 °C) and the intercepts from these regressions were used to calculate daily RMR on a minute-by-minute basis for current Ta’s, and to estimate the effect of climate change (from 5 °C below, to 3 °C above current Ta’s) on the projected RMR for high- and low-altitude sloths.

Results

Resting metabolic rate (temperature manipulation in the metabolic chamber)

Mean body mass across the 12 sloths was 5.33 ± 0.67 kg (SD) and mean RMR over all temperatures was 118.26 ±  36.76 kJ/kg/day (Table S2). Mean RMR values were 39% lower than the general mammalian allometric prediction of Kleiber (1961), and 13% lower than the prediction of White & Seymour (2003) which includes variation due to factors such as body temperature and digestive state. Neither body mass (p = 0.85) nor sex (p = 0.73) had a significant effect/interaction on RMR (Table 1). There was no significant difference in RMR for sloths that had (121.59 kJ/kg/day ± 17.21 kJ/kg/day) and had not (117.58 kJ/kg/day ± 39.59 kJ/kg/day) been sedated prior to entering the metabolic chamber (w = 939, p = 0.329).

The LMMs showed that ambient temperature (represented as a continuous variable in LMM1 [t = 8.70, p < 0.001]) had a significant effect on RMR (Table 1) (Fig. 1, Fig. S3A). There was also a significant interaction between ambient temperature (represented as a categorical variable in LMM2) and altitude on RMR (t = 3.72, p < 0.001) and a significant effect of body temperature (t = 5.22, p < 0.001) on RMR (Table 1) (Fig. 1, Fig. S3B). A significant effect and interaction of body temperature on RMR can be seen in Fig. S3C. As there were multiple measurements taken from each individual, individual effects were accounted for in the model using estimates of the random effect (Fig. S4).

RMR of high-altitude sloths (n = 4) (126.25 ±  40.84 kJ/kg/day) was significantly higher than RMR of low-altitude sloths (n = 8) (110.70 ±  30.94 kJ/kg/day) when all the data were considered together (w = 1763, p = 0.038) (Fig. S3A). At Ta’s ≥32 °C, RMR values of high-altitude sloths (n = 4) (162.71 ±  52.03 kJ/kg/day) were significantly higher than those of low-altitude sloths (n = 5) (103.72 ±  34.69 kJ/kg/day) (t = 3.08, df = 17.52, p = 0.007) (Fig. S3B). There was no significant difference in RMR between high- and low-altitude sloths at Ta’s <32 °C (w = 1013, p = 0.44). Metabolic rates were lowest at 16 °C–19 °C (high-altitude: 90.90 ±  23.75 kJ/kg/day, low-altitude: 83.53 ±  21.74 kJ/kg/day) and increased with increasing Tabefore plateauing at temperatures between 23 °C–32 °C (high-altitude: 134.19 ±  27.42 kJ/kg/day, low-altitude: 127.21 ±  24.73 kJ/kg/day). At Ta’s above 32 °C, high-altitude sloth RMR increased sharply, while low-altitude sloth RMR decreased (Fig. S3 B). In high-altitude sloths, Tb at Ta’s ≥32 °C (35.87 ±  0.76) was significantly higher than Tb at Ta’s <32 °C (34.51 ±  0.73) (t = 5.31, df = 15.23, p < 0.001).

The effect of natural fluctuations in Ta on Tb and posture

There was a significant effect of Ta on Tb (F = 25.98, p < 0.001) (Table 2), and a significant effect of altitude origin on Tb (t =  − 44.95, p < 0.001) (Fig. 2). There was a significant effect of Julian day (t =  − 7.11, p < 0.001) and time of day (F = 81.99, p < 0.001) on Tb (Table 2) after controlling for temporal autocorrelation. The relationship between Ta and Tbwas best described using a linear model (delta AICc = 0; polynomial: delta AICc = 0.58) (Fig. 2). Mean Tawithin the enclosures was 26.90 °C ± 1.93 °C (overall recorded minimum: 24.56 °C, maximum: 33.11 °C). Rectal temperatures averaged 34.84 °C ± 0.88 °C across individuals, ranging from an overall recorded minimum of 33.43 °C to a maximum of 37.28 °C. The mean Tb range within each individual was 2.60 °C. There was a mean lag of 2.5 h between the maximum/minimum daily ambient temperature and the maximum/minimum sloth Tb(Fig. 3). Tb of high-altitude sloths (35.70 ±  0.61 °C) (n = 4) was significantly higher than that of low-altitude sloths (n = 8) (34.34 ±  0.56 °C) (w = 19604, p < 0.001) (Figs. 1 and 2). Ta (F = 10.33, p < 0.001) and time of day (F = 1.54, p = 0.01) had a significant effect on sloth body posture. Tb also had a significant effect on sloth posture with both high- and low-altitude sloths adopting spread out postures more frequently at higher temperatures (Table 2) (F = 4.30, p = 0.04); however, model selection showed that GAMM2 with Ta was the best fit model (delta AICc = 0; GAMM3: delta AICc = 1.38).

Figure 1 The effect of ambient temperature (Ta) on the resting metabolic rate (RMR) and body temperature (Tb) of Choloepus hoffmanni sloths originating from high and low altitudes.

Means presented (+ SD) are taken from 12 animals (4 high altitude, 8 low altitude). Ta significantly affected RMR for both high and low altitude sloths. Both high and low altitude sloth Tb were significantly affected by changes in Ta.

Projected impacts of climate change

There was a projected Tb increase of 1.53 °C and 2.13 °C, respectively, for high-altitude and low-altitude sloths if the climate warmed by an average of 2 °C (Fig. 4). For low-altitude sloths, the mean projected Tb was 36.57 ±  0.01 °C (95% CI [36.564 °C–36.576 °C]) indicating high precision in the estimates due to the larger sample size (n = 25). For high-altitude sloths, the mean projected Tb was 37.27 ± 0.01 °C (95% CI [37.258 ° C–37.283 °C]) reflecting greater variability and less precision due to the smaller sample size (n = 3) (Fig. S2). As climate change causes an increase in average daily Ta, the RMR of both high- and low-altitude sloths is projected to increase accordingly (Fig. 5). As the increase in average daily Ta exceeds 2 °C above current Ta’s, low-altitude sloth RMR is projected to plateau, while high-altitude sloth RMR continues to escalate (Fig. 5). The 95% confidence intervals (Table S3) reflect the greater variability in the estimates for high altitude sloths due to the smaller sample size.

Table 2 Results of GAMM1 describing the effects of Ta, altitude and other covariates on Tb. Results of GAMM2 and GAMM3 describing the effects of Ta and Tb and posture, respectively.

Statistical test (dependent variable)		SE	t-value	F	p-value	
Predictor variable					
GAMM1 (Tb)			
Ta (natural)				25.98	0.001	
Altitude		0.03	−44.95		0.001	
Julian day		0.05	−7.11		0.001	
Time of day				81.99	0.001	
GAMM2 (posture)			
Ta (natural)				10.33	0.001	
Time of day				1.54	0.013	
GAMM3 (posture)			
Tb				4.30	0.04	
Time of day				6.04	0.001	

Figure 2 The effect of Ta and altitude origin on Tb.

There was a significant effect of ambient temperature (F = 25.98, p < 0.001) and altitude origin (t =  − 44.95, p < 0.001) on sloth body temperature, with high-altitude sloths (red) having significantly higher body temperatures compared to low-altitude sloths (blue) across the range of ambient temperatures. The shaded area represents 95% confidence intervals.

Figure 3 Natural fluctuations in Ta and Tb of Choloepus hoffmanni sloths over time.

The solid line shows the mean Tb of 28 animals. Standard error was typically 0.17 (error bars too small to plot). There was a mean lag of 2.5 h between the maximum/minimum daily ambient temperature and the maximum/minimum sloth Tb.

Figure 4 The projected impact of climate change on the Tb of highland and lowland Choloepus hoffmanni sloths.

Due to the limited ability of sloths to metabolically regulate Tb in response to temperature variation, if climate change were to cause a 2 °C increase in Ta, the highland sloth Tb is predicted to increase by 1.53 °C while lowland sloths will experience a Tb increase of 2.13 °C. Data modelled over several days until equilibrium using data from 28 sloths.

Figure 5 The projected impact of climate change on the RMR of Choloepus hoffmanni sloths originating from high and low altitude forests.

Modelled from 5 °C below to 3 °C above current Ta’s. Error bars represent the confidence intervals based on bootstrap analysis adjusted for sample size. As climate change increases average daily Ta, the RMR of both high and low altitude sloths is projected to increase accordingly. As the increase in average daily Ta exceeds 2 °C above current Ta’s, however, the capacity of low altitude sloths to invoke metabolic depression halts any further increase in RMR. Sloths originating from highland forests are projected to experience a continuing escalation in metabolic rate.

Discussion

The sloth RMR data are similar to those values measured previously for sloths in both the Bradypus and Choloepus genera (Cliffe et al., 2018; Lemaire et al., 1969; McNab, 1978; Vendl et al., 2016), lending support to the notion that all sloths have a metabolic rate which falls far below the value expected for a mammal of similar size (Irving, Scholander & Grinnell, 1942). Specifically, sloth RMR values were found to be 39% lower than the general mammalian allometric prediction of Kleiber (1961), and 13% lower than the prediction by White & Seymour (2003), which incorporates adjustments for factors such as body temperature and digestive state. This closer alignment with the White & Seymour model is consistent with the sloth’s unique physiological traits, including their low and variable body temperature and slow digestive rate.

The reduced metabolic rate of sloths has been linked to reduced thyroid activity (Lemaire et al., 1969) and a low caloric intake combined with long digesta retention times, restricting the rate at which energy can be acquired (Cliffe et al., 2015; McNab, 1978; Montgomery & Sunquist, 1978; Nagy & Montgomery, 1980). A manifestation of this is in the field metabolic rate (FMR), the energy expenditure of a free-living animal in the wild (Nagy, 1987), which is typically about three times higher than the resting rate in normal mammals (Fei et al., 2016; Withers, 1951). In contrast, sloth FMR is only 1.3 times higher than sloth RMR (Pauli et al., 2016), which is likely to be attributable to the low levels of sloth activity at all times. As part of this reduced metabolic rate strategy, all sloths appear to operate at a lower and more variable body temperature than most mammals (Britton & Atkinson, 1938; Irving, Scholander & Grinnell, 1942; Montgomery & Sunquist, 1978). Indeed, the mean Tb, maximum Tb, and overall Tb range we recorded for each individual were within 3% of the corresponding values reported for wild sloths (Pauli et al., 2016). However, despite the co-varying ambient temperatures and body temperatures of sloths (Fig. 3), there are metabolic consequences of temperature variation.

At mid-low Ta’s, both high- and low-altitude Choloepus sloths showed a similar metabolic response to variation in temperature to that observed for the Bradypus genus (Cliffe et al., 2018). At lower temperatures, this comprises an increase in metabolic activity with temperature in a manner similar to ectotherms. Given the sloth’s marked plasticity in Tb, this is likely a passive effect of increased temperature on the rate of enzymatic reactions within the body (Daniel et al., 2010). The increase in RMR eventually results in a metabolic plateau at Ta’s which coincide with the typical range of ambient conditions in tropical forests (23–32 °C) (Cliffe et al., 2023; Giné et al., 2015). This metabolic plateau (or nominal Choloepus ‘thermally-active zone’) spans a broader range of Ta’s than that observed for the Bradypus (26–30 °C), and may underlie the comparatively broader geographic range of Choloepus sloths (Montgomery & Sunquist, 1978; Pauli et al., 2016).

The most notable finding from this work, however, is the stark difference in RMR between high- and low-altitude sloths when Ta’s rise above 32 °C. In these conditions, low-altitude animals appear to depress their metabolic activity in a manner that is comparable to the Bradypus sloths (Cliffe et al., 2018), without entering into a state of torpor, hibernation, or aestivation. Sloths of the same species originating from high-altitude regions, however, appear to be unable to modulate metabolic rate in this way, with RMR increasing at temperatures above 32 °C. This continued increase in metabolic rate may simply represent a broader thermal window for high-altitude sloths which would perhaps be expected for an animal originating from a more variable thermal environment (Rohr et al., 2018; Shokri et al., 2022; Sun et al., 2022). However, the corresponding significant increase in body temperature for these animals at temperatures above 32 °C suggests that the contrasting metabolic response between sloths from different altitudes is more likely a metabolic adaptation to climatic differences (Norin & Metcalfe, 2019).

Mid-day temperatures in lowland tropical forests frequently rise well above 30 °C (Aguilar et al., 2005), and, aside from some nominal utilisation of microclimates within the canopy, sloths, have little ability to escape the heat (Britton & Atkinson, 1938; Montgomery & Sunquist, 1978). In such conditions, an ability to invoke metabolic depression would reduce metabolic heat production and therefore minimise both Tb and energy expenditure.

The regions from which the high-altitude sloths used in this study originate (>1,000 m above sea level) are typically 3−9 °C cooler than the corresponding lowland forests (Pounds, Fogden & Campbell, 1999), and the sloths living at altitude are adapted to the colder climate with darker colouration and longer, thicker fur (Enders, 1940; McNab, 1985). This difference in pelage is likely to minimise their thermal conductance, and buffer them against fluctuations in Ta. The result is reflected in the consistently higher Tbof high-altitude sloths compared to those from lowland regions at a given Ta (Figs. 1 and 2). In tandem with this, high-altitude sloths also maintain an overall higher RMR than their lowland counterparts at the same Ta, which presumably enables them to survive in a colder climate (Anderson & Jetz, 2005; Haim & Izhaki, 1993; McNab, 2002; Zhao et al., 2014). As the Ta in highland forests rarely exceeds 30 °C (Pounds, Fogden & Campbell, 1999), sloths there should have little need to invoke metabolic depression in response to high temperatures, and consequently it appears that these animals do not have the ability to do so. Interestingly, this is in stark contrast to earlier findings in rodents, where golden spiny mice living by the Dead Sea, which is always warm, cannot up-regulate their RMR, while those from Mount Sinai can (Haim & Borut, 1981). The difference in metabolic response between sloths of the same species originating from different altitudes likely reflects distinct reaction norms shaped by genetic variation and environmental influences (Pettersen & Metcalfe, 2024). This many include early-life conditions and developmental plasticity, as temperature during embryonic development and early growth stages can influence metabolic and thermoregulatory mechanisms (Pettersen & Metcalfe, 2024; Schnurr, Yin & Scott, 2014; Scott & Johnston, 2012).

The precise molecular mechanisms involved in the active depression of metabolic rate in mammals are poorly understood and likely to be multi-faceted (Andrews, 2019; Carey, Andrews & Martin, 2003; Giroud et al., 2021; Levesque, Nowack & Stawski, 2016; Rider, 2016; Storey, Heldmaier & Rider, 2010). The initial metabolic suppression seen in mammals entering hibernation, which precedes any drop in Tb, isthought to be partially triggered by reversible changes in gene expression (Hittel & Storey, 2002). However, the depression of sloth metabolism in response to high Ta’s occurs at a faster rate than transcription or translation can probably occur (Staples, 2014). In such cases, current evidence points towards a mechanism of active suppression in mitochondrial metabolism through the regulation and activation of pre-existing proteins as a driver for rapid changes in mammalian metabolic activity (Rider, 2016; Staples, 2014).

While the results reported here should be considered preliminary due to the acute temperature changes tested and sample size limitations, it is clear that animals from the Choloepus genus originating from different altitudes respond metabolically in different ways when faced with high ambient temperatures.

Projected impacts of climate change

How organisms obtain, convert and expend energy is directly related to the Ta of their environment (Brown et al., 2004; Levy et al., 2017) and this is one of the reasons why climate change is projected to have an extensive effect on the global energetic requirements of organisms (Dillon, Wang & Huey, 2010; Parmesan, 2006; Parmesan & Yohe, 2003; Root et al., 2003; Shokri et al., 2022). What might the consequences be for sloths?

While future climatic predictions for the South and Central American rainforests are variable, all point towards these regions becoming hotter and drier, with current estimates forecasting a 2–6 °C increase in average daily air temperatures by the year 2100 (Marengo et al., 2014; Nũez, Solman & Cabré, 2009; Romero & J, 2022). From the data presented in this paper, we were able to create a simplistic model to predict the possible effect of climate change-associated temperature increases on the Tb and RMR of sloths originating from both high- and low-altitude regions.

As climate change causes an increase in average daily Ta, the RMR of both high- and low-altitude sloths is projected to increase accordingly. As the increase in average daily Ta exceeds 2 °C above current Ta’s, the capacity of low-altitude sloths to invoke metabolic depression limits any further increase in RMR (Angilletta, 2009; Dillon, Wang & Huey, 2010). This physiological plasticity should be accentuated by the ability of lowland populations to shift distribution ranges along climatic gradients to higher elevations (Parmesan & Yohe, 2003; Perry et al., 2005; Root et al., 2003), thereby providing a degree of flexibility when faced with a warming climate. On the other hand, sloths originating from high-altitude mountain-top locations appear to lack the metabolic and geographic plasticity of their lowland counterparts and consequently are likely to be more constrained in their ability to adapt to a continuously warming climate.

An increased rate of energy expenditure must be balanced by an increased rate of energy intake. This option appears biologically implausible for sloths due to their slow digestive rate and constantly full stomach, restricting food intake and imposing intrinsic energy processing limitations (Cliffe et al., 2015; Montgomery & Sunquist, 1978; Nagy & Montgomery, 1980). Indeed, estimates for digesta passage time for sloths range from 150–1,200 h (Foley, Engelhardt & Charles-Dominique, 1995; Montgomery & Sunquist, 1978; Vendl et al., 2016), some 3–24 times slower than similar sized arboreal folivores (Espinosa-Gómez et al., 2013), with the primary reason for this believed to be linked to the time required to detoxify the food plants (McNab, 1978). While it is plausible that the increase in metabolic activity with environmental temperature may increase the rate of food passage (Doucette et al., 2023), and therefore intake, it is unlikely that the sloth’s digestive tract has the capacity to process food much faster. Although the model presented here is rudimentary in its omission of error and uncertainty considerations, and further research is needed to fully understand the sloth’s metabolic response to temperature, we predict that a comparatively small increase in ambient temperature could see high-altitude sloths pushed into a situation where it is impossible to make their energy consumption tie in with their energy budget.

Supplemental Information

Supplemental Information 1 Origin location, altitude, and climate information for the 12 C. hoffmanni sloths used for metabolic testing

Supplemental Information 2 Altitude origin, body mass, resting metabolic rate (RMR), allometric predictions and body temperature (Tb) data for 12 C. hoffmanni sloths at different ambient temperatures (Ta)

Supplemental Information 3 RMR projections for high- and low-altitude sloths under different climate change scenarios

The confidence intervals (CI) were calculated using a bootstrap approach adjusted for sample size, ensuring accurate representation of variability.

Supplemental Information 4 Residual diagnostic plot from the LMM looking at the effects and interactions of ambient temperature, body temperature, altitude, sex, and body weight on RMR

Supplemental Information 5 Density plots of projected body temperature ( °C) for sloths under future climate scenario

Predictions based on a 2 °C increase in ambient temperature. The top plot shows the distribution of projected body temperatures for low-altitude sloths, while the bottom plot shows the distribution for high-altitude sloths. The red dashed lines indicate the 2.5th percentiles, and the green dashed lines indicate the 97.5th percentiles of the projected temperatures, illustrating the range of uncertainty around the predictions. The density plots are derived from 1,000 bootstrap simulations to account for variability and uncertainty in the model predictions.

Supplemental Information 6 The regression relationship between ambient temperature and RMR in high (red) and low (blue) altitude sloths, representing the interaction between altitude and ambient temperature on RMR

(A) Derived from LMM1, depicting ambient temperature on a continuous scale (the shaded areas represent 95% confidence intervals). (B) Derived from LMM2, depicting ambient temperature on a categorical scale. (C) The regression relationship between body temperature and RMR in high (red) and low (blue) altitude sloths, representing the interaction between altitude and body temperature on RMR (the shaded areas represent 95% confidence intervals).

Supplemental Information 7 Plot from LMM2 of the random effect estimates from Animal ID for each sloth

The intercept represents the population mean RMR, with blue dots representing the mean distance from the population mean for each individual, and black lines representing the variation in RMR for each individual.

Supplemental Information 8 Raw body temperature data for Choloepus Hoffmanni sloths

Supplemental Information 9 Raw data for Choloepus Hoffmanni resting metabolic rate

Supplemental Information 10 ARRIVE 2.0 checklist

We thank the Sloth Sanctuary of Costa Rica for allowing us to conduct this research on their property and their advice, and Dr. Francisco Arroyo for his veterinary and logistical assistance throughout data collection.

Additional Information and Declarations

Competing Interests

Author Contributions

Animal Ethics

Data Availability

The authors declare there are no competing interests. The authors are not aware of any competing interests that the Indiegogo crowdfunders and Sloth Conservation Foundation donors may have.

Rebecca N. Cliffe conceived and designed the experiments, performed the experiments, analyzed the data, prepared figures and/or tables, authored or reviewed drafts of the article, and approved the final draft.

Heather E. Ewart analyzed the data, prepared figures and/or tables, authored or reviewed drafts of the article, and approved the final draft.

David M. Scantlebury conceived and designed the experiments, performed the experiments, analyzed the data, authored or reviewed drafts of the article, and approved the final draft.

Sarah Kennedy performed the experiments, authored or reviewed drafts of the article, and approved the final draft.

Judy Avey-Arroyo performed the experiments, authored or reviewed drafts of the article, and approved the final draft.

Daniel Mindich performed the experiments, authored or reviewed drafts of the article, and approved the final draft.

Rory P. Wilson conceived and designed the experiments, performed the experiments, analyzed the data, authored or reviewed drafts of the article, and approved the final draft.

The following information was supplied relating to ethical approvals (i.e., approving body and any reference numbers):

This research was approved by the Swansea University Animal Welfare & Ethical Review Process Group (AWERP), and the Costa Rican government and associated departments (MINAE, SINAC, ACLAC) permit number: R-033-2015. All research was performed in accordance with relevant guidelines and regulations.

The following information was supplied regarding data availability:

The raw data on sloth metabolic rate versus temperature are available in the Supplementary File.

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
