# Peer review of "Sloth metabolism may make survival untenable under climate change scenarios"

_PeerJ, doi:10.7717/peerj.18168_

## Round 0.1 · original submission · Major Revisions

Dear Dr Cliffe and colleagues,
Many thanks for submitting your manuscript to PeerJ.
The manuscript has been reviewed by two reviewers. Following these comments, along with my own reading of the paper, I am recommending major revisions.

I enjoyed reading the paper and both reviewers highlight its merit and the timely importance of the findings, but they also point out some major issues that need to be addressed before the manuscript can be considered for publication. In addition to reviewers' comments, I would like to add the following points:

1) in line with one of reviewer 2's comments, I would like to see more details on how the LMM models are constructed. For instance, in line 246, you say that captive status was included as covariate but it is not immediately clear to me what you mean by this since all animals are captive. You also test the difference in RMR between high vs low altitude sloths using a Mann-Whitney U test. I would recommend, instead, to include altitude status (high vs low) in the LMM model along with temperature, sex and body mass to assess their effect on RMR. This would also take into account pseudoreplication, which the Mann-Whitney U test does not. As a side note, unlike what you state in line 273-274, the results of the Mann-Whitney test do NOT show a significant difference in RMR for temperature > 32 degrees since p = 0.051.

2) The main hypothesis, namely that altitude origins (high vs low) predict sloths' metabolic responses to changes in temperature, is not actually tested statistically. I would suggest to, potentially, split temperature values into meaningful range values and then test the effect of the interaction between these categorical temperature values and altitude origins (high vs low) on RMR

3) for all LMM analyses, please confirm that they meet the basic assumptions (e.g., normality of residuals, homoscedasticity etc...)

Reviewer 1 ·

Basic reporting

The authors have conducted an experiment on the resting metabolic rate of a species of sloth across a range of temperatures, comparing the response of the resting metabolic rate to varying temperatures across two populations: high altitude and low altitude. Overall, I found the findings of the study to be timely and interesting. However, I have some general comments that need to be addressed.

Experimental design

There is scant detail on the local annual/seasonal climate profile of the two conspecific species, including the conditions they naturally experience. Furthermore, I did not find details on the rationale behind the selection of assessment temperature levels. I also suggest adding information on the altitudes of the specimens' origins, considered here as low and high altitude.
Secondly, the exposure of the specimens to temperature changes is acute, which must be stated. Generally, to avoid thermal shock, a specimen's metabolic rate is assessed after a certain period—at least several days—for acclimation or acclimatization. While I understand that this may be challenging to carry out with this species, I suggest the authors state that the observed response of metabolic rate to acute temperature changes is preliminary. Therefore, caution must be exercised in discussing the results, as the animals could require a longer period to potentially adjust their metabolic responses.
Thirdly, I did not find the summary of the linear mixed model in the manuscript.

Validity of the findings

Specific comments:
Line 399-401: Why did the authors interpret the continued increase of metabolic rate with temperature as a lack of metabolic plasticity? This could be because specimens at higher altitudes have a broader thermal window.
Line 414-416: I found this to be a very broad conclusive sentence, beyond the findings of this study. To confirm such a conclusion, further studies are needed.
Overall, I believe that while the study has potential, there is substantial room for improvement in both the presentation and interpretation of the results. In the meantime, I recommend keeping the following reference in mind concerning metabolic responses to temperature, which might also help in discussing the findings: Shokri, M., et al. (2022). Metabolic rate and climate change across latitudes: Evidence of mass-dependent responses in aquatic amphipods. Journal of Experimental Biology, 225(22), jeb244842.

·

Basic reporting

Overall comments.
This manuscript presents a study on the resting metabolic rate (measured in metabolic chambers) and body temperature (measured rectally) of highland and lowland two-fingered sloths (Choloepus hoffmanni). Overall, the manuscript is well-written, with clear English language and few (if any) grammatical errors. The authors present resting metabolic rates (RMR) and body temperatures (Tb) responses to ambient temperature (Ta) and show that sloths originating from highland regions appear to be more sensitive to high Ta than sloths originating from lowland regions. The authors use their empirical data to generate projections of RMR and Tb under future temperature regimes, to show that highland sloths are likely to be more vulnerable to climate change.

Overall, the study is interesting and provides new insights into potential challenges that sloths, which are poikilothermic, may face in a warming climate. The authors have done a good job of describing the study, but there are some major weaknesses which should be addressed prior to acceptance, particularly concerning the statistical analyses. I present these comments below, followed by detailed, in-line comments on the different sections.

Raw data.
Thank you for providing the raw data, however this file needs more descriptive metadata to be useful to a reviewer or others. There are 28 sets of columns with the titles “Date / time”,”Temp”, “Activity”, “Arms” with no explanation to them. Are these temperatures recorded from the sloths in enclosures? In which case they should be in a separate sheet/file than the RMR measurements. An additional sheet or a readme-file with explanations for each column would also be very useful. The part of the data that is from the metabolic chambers appear to be means per ‘experimental run’, but this is not clearly stated. Why not present the raw measurements?

ARRIVE checklist.
The checklist appears to be complete, and well detailed with references to sections in the manuscript.

Thank you and kind regards,
Dr. Monica Trondrud

Experimental design

The experimental design and metabolic chamber study is well explained, although it seems the authors have made use of opportunistically available sloths from a sanctuary, resulting in rather skewed sample sizes for the analyses of Tb under naturally fluctuating Ta. Further, the projected impacts stem from a rather simplistic prediction which is not sufficiently explained and does not seem to account for uncertainties/error in the data.

Validity of the findings

The major weakness of the manuscript concerns the explanation for the statistical analyses, how the results from these are presented, and the predictive modelling presented in the discussion. First of all, it is not clear from the methods on lines 244-249 whether you actually fitted multiple models or just one, because you do not explain the difference between the models that were tested. Here, I would like to have seen tables (e.g. in your supplementary material) showing the structure of each model and the model selection process (with results for each step), for all analyses that are presented in the results. Did you use stepwise model selection (as suggested since you use ANOVA), or did you fit different models for each covariate and compared using AICc? When using data including random effects, it is important to fit models using maximum likelihood (ML) during model selection steps, while the default is to use restricted maximum likelihood (REML) for the final model(s), to account for the random effects. Did you change these during the model selection and when presenting the final model(s)?

Further, it is not clear to me how you fitted the raw/processed data in the model(s). Table S1 shows that some sloths had up to 16 measurements taken, while others only 1 measurement. Did you use one mean value from each sloth per ’experimental run’? Why did you not use the raw measurements from each ‘experimental run’?

With regards to the analyses of Tb in response to Ta for the sloths in enclosures, it is not clear to me how you accounted for individual random effects or the cyclicity and temporal autocorrelation in the temperature measurements. From what is written on lines 256-259, you state “The relationship between ambient temperature and body temperature was examined using a linear model and the mean time lag between ambient and core body temperature was determined by calculating the average time between maximum / minimum ambient temperature and maximum / minimum core body temperature for each individual.”
First of all, if you have used data from multiple individuals, you should have used a mixed model and accounted for individual random effects. Second, ambient temperature (and body temperature) both seem to follow a cyclical pattern from day to night (figure 2). How did you account for time in these models?

Temporal autocorrelation is a common characteristic of temperature data, and especially for ambient temperature. I would suggest that you try and fit a gamm instead, and test whether the relationship between Ta and Tb, and Tb and time of day, are truly linear. If you fit Tb with a thin-plate regression spline and time of day as a cubic circular spline, you can look at the model plots and evaluate whether these result in linear relationships or not. For instance, using the gamm function from package mgcv or gamm4 your model could look like this (this is only a suggestion):

fitted.model <- mgcv::gamm(Tb ~s(Ta)+s(time of day, bs=”cc”)+calendar day/season+s(id, bs=”re”), data=data)
and then you can test the temporal autocorrelation using
itsadug::start.value.rho(fitted.model, plot=T) to evaluate the autocorrelation in your Tb data. If this is autocorrelated, you need to refit the model with a corAR or ARIMA function.

My last comment with respect to this part of the results is that the figure legend for figure 2 claims to present “the effect of Ta on Tb”, but in reality, this only shows how Tb and Ta vary over time. Since you are fitting regressions to all of your data, it would be good if you actually plotted the fitted data from these regressions (including confidence intervals for the predictions, not SE) on top of adjusted or raw data.

“Projected impacts of climate change”

The predictive modelling of future climate responses should be explained as part of the methods under “statistical analyses”, and the results of the predictions should be presented as part of your results. The discussion that follows the outcome of these predictions is the only part that should be presented in the discussion.

Secondly, the method described to estimate these projected impacts is not very detailed and the resulting projections appear to simply be an adjustment of the intercepts in your models. Considering that you have data from multiple individuals and measurements, you are bound to have a certain amount of error around each estimate. This error should be included in your projections, preferably using a bootstrap method where you generate predictions including standard errors drawn from a normal distribution using a mean of 0 and a standard deviation equal to the standard error of your model predictions. If you e.g. simulate 1000 predictions per Ta measurement, you can include uncertainty around your projected responses, which will likely make them more conservative but also more trustworthy.

Additional comments

Line 42. Perhaps include a bit more details/examples to this statement?

Line 46. I would appreciate if you can place the relevant reference immediately following each of the three examples of adaptation.

Line 55. Except that in some cases of animals who are well adapted to heat, they can allow body temperature to rise by a few degrees to avoid energy-costly responses (see e.g. papers by Fuller et al., Mitchell et al)

Line 72. Is “homeothermy” truly the “norm” for mammals? Even large ungulates can have fluctuations in body temperatures, see e.g. a brief summary of examples in part 6 of the following publication: Mitchell D, Snelling EP, Hetem RS, Maloney SK, Strauss WM, Fuller A. Revisiting concepts of thermal physiology: Predicting responses of mammals to climate change. J Anim Ecol. 2018; 87: 956–973. https://doi.org/10.1111/1365-2656.12818

And, for instance:

Danielle L Levesque, Ana M Breit, Eric Brown, Julia Nowack, Shaun Welman, Non-Torpid Heterothermy in Mammals: Another Category along the Homeothermy–Hibernation Continuum, Integrative and Comparative Biology, Volume 63, Issue 5, November 2023, Pages 1039–1048, https://doi.org/10.1093/icb/icad094

Line 92. The statement “[…] which may change how these high-altitude populations fare against a warming climate […]” is vague, and it would be good if you could expand on the specific mechanism by which you expect insulation to have metabolic and Tb implications.

Lines 95-101. Seeing as you have conducted a controlled comparison between two groups, it would be good if you could state some specific predictions to your results. E.g. whether you expect responses to be the same, or different between the two groups, and why.

Methods

There are no references to the supplementary material describing the results concerning effects of anesthesia (which, in any case are rather weak considering you only have two individuals in this group).

Line 115. What were the criteria for the sloths to be “deemed suitable for participation”?

Line 116. Please provide regions and altitude range of these regions from where the sloths originated.

Line 123. Please provide the sex of the two controls.

Line 126. Were the loggers coated in inert wax or some other kind of surgical/biological coating?

Line 130. Why only every 30 minutes? If each adjustment period took 60 mins you only have 2 body temps for each adjustment period, and for each RMR record you may not have a matching Tb.

Line 187. Here you have some repetition of references at the end of the sentence.

Line 203. 20°C variation is not necessarily extreme depending on the climate and species. Do you mean that this is the absolute range of temperatures these sloths are naturally exposed to? Perhaps writing “most extreme” makes it easier to understand that you refer to the maximum range for the sloths, and not an extreme range in general?

Line 246. Weren’t all sloths used in the RMR study in captivity? What is defined as captivity status?

Line 247. How many models did you test? How did these differ from each other? The text suggests you fitted one LMM with RMR as the response, and ambient temperature, body mass, sex, and captivity status as covariates, and ID as a random factor. What were the different models?

Line 247. Why did you not test the relationship between Tb and RMR?

Line 256. It is not clear here whether you used multiple individual measurements of Tb and if so, how you accounted for repeated IDs or temporal autocorrelation in your models. And why did you not include sex, highland/lowland and body mass in the models of Tb?

There is no explanation of the method used to make the predictions of how climate change will affect Tb in highland and lowland sloths.

Results

Overall, there is an extensive and variable use of decimals for your p-values and chi-squared tests values. I would keep to max. 2 decimals and present any p-value lower than 0.01 as P<0.01 lower than 0.001 as P<0.001 and so on, and not as “p= 1.27e-07”(for example).

Line 264. The title of this first paragraph is misleading because it says “effect of […] altitude”, but the measurements were not conducted at different altitudes, were they? I would use the same phrasing as in the title of the next paragraph: “Effects of body mass and sex on RMR (metabolic chamber)”. In my suggestion I have removed “for high and low altitude sloths” because you state that there were no differences between the sexes or between body masses, and I presume that you did not/cannot find differences within low- and high-altitude individuals given the relative low sample sizes in each group.

Lines 270-274. It is unclear to me why these sentences are not placed in the paragraph below titled “Effects of Ta on RMR, body temperature and body posture for high and low altitude sloths (metabolic chamber)”

Line 284. r2 is normally presented with capital R if it refers to the R2 of a regression model.

Discussion

Overall, your discussion could benefit from some consideration of the limitations of your results given the skewed sample sizes for the sloths in enclosures and relatively small sample sizes for the metabolic chamber study.

Line 324. But you do not show the relationship between Tb and RMR, which, if positive and significant, can support this statement.

Line 353. These are sloths from the same species, so how do you suggest that this is ability is lost, or never gained, the highland individuals? Could it be a reaction norm of the environment, or a result of early-life condition? Please include some references and more detailed reflecting to strengthen this argument.

Line 378 and onwards are methods and results and should be presented in their respective sections.

Figures

Figure 2 does not show “The effect of natural fluctuations in Ta on the Tb of Choloepus hoffmanni sloths”. This figure shows the simultaneous fluctuation in Tb and Ta over time. You need to plot Tb as a function of Ta to show the effect of Ta on Tb.

For figures 3 and 4 it would be better if you generated predictions from a more robust bootstrapping method to include uncertainty around them, especially considering that you have relatively skewed sample sizes in the Tb data for low- vs highland sloths, and presumably there is more uncertainty around the estimates for the highland sloths.

---

## Round 0.2 · Minor Revisions

Dear Dr Cliffe and colleagues,

Many thanks for revising your manuscript. I sent your revision back to the original reviewers who are both happy with the way you addressed their original comments. Following this, I am in principle happy to accept your manuscript for publication provided that you address some additional minor comments one of the reviewers has made, especially in the methods and result sections.

Reviewer 1 ·

Basic reporting

I thank the authors for addressing my earlier comments. Having read the revised version, I found it improved and have no further comments/ suggestions.

Experimental design

no comment

Validity of the findings

no comment

·

Basic reporting

The authors have done a tremendous job in revising the manuscript and accommodating the reviewers and editor’s suggestions. The manuscript has improved substantially. I appreciate the effort the authors have made, and while I still have some comments, these are mainly requesting further clarifications.
Most importantly, I was not able to find Tables 1 and 2 (which I believe should include the model outputs) in the manuscript (neither the .docx file or the .pdf) or in the supplementary material.
Thank you for re-organising the raw data.
Regarding the sample sizes, I did not mean to invalidate your work, but meant to highlight the uncertainty in the projections based on the small sample size for high-altitude sloths. I appreciate that you now comment on this uncertainty, both in the results, and in the discussion. I recognize the challenges working with such a difficult species, so I apologize for the bluntness of my previous comment.

The only major issue is the lack of tables 1 and 2. In addition, there is a need for a reorganising of the figures (presented in the wrong order), and checking of your references (many are duplicated). I also only request some clarifications and improvements to the description of results and methods (see specific comments below).

Experimental design

No comment

Validity of the findings

It is unclear to me why you calculated the expected allometric relationships for RMR when you do not mention these in the discussion. It is perhaps interesting that there is a better correspondence with the calculations that account for body temperature and digestive states, but there is no acknowledgement of this in your discussion and I wonder if you need to include them? The results are perfectly valid without this comparison. Alternatively, perhaps you can include a few sentences in your discussion adressing these differences?

Otherwise, I believe your findings and conclusion based on your findings are sound, and slighty milder in their wording which better reflect the data you have at hand.

Additional comments

Line 118: perhaps you could move the coordinates to immediately after “Sloth Sanctuary of Costa Rica” instead of at the end of the sentence?

Line 121: thank you for providing the inclusion criteria, this is now much clearer

Line 242: a small detail, but please provide version you used in R (the last version used). This can be relevant for the reproducibility of your analyses.

Line 246: Many authors reference the R package creators/authors, to acknowledge their work. If your reference list is not already maxed, I would prefer to see these included, but it is not the end of the world.

Line 261: The sentence “Final models also discounted any non-significant effects and interactions” seems to me like a backwards stepwise model selection approach (in which you should test the removal of each variable using e.g. ANOVA), which normally are based on log-likelihood criteria (not AICc). It therefore seems to me that you have used two different model selection criteria for your models, but it is not clear how (or why) you did this, as it is more common to only use one model selection approach (stepwise model selection or AIC). Please clarify this section to explain whether:

a) You had two full models (all variables, all data) with only Ta differing between a continuous and a categorical value, which where then compared with AICc and then the best ranked model (lowest AICc) was subsequently selected with stepwise backwards model selection (removing one explanatory variable/interaction at a time and testing their significance with an F-test or t-test, e.g. using ANOVA)

or

b) You first used stepwise model selection on each of the two models (with either Ta as categorical or continuous) then compared the final models, with only variables that were statistically significant, using AICc

Also, please include the AICc selection criteria, (i.e. selecting the model with lowest AICc score and what you did if both models were within a given delta AICc, normally either 2 or 5)

Line 274: Unclear why you fit body temperature as a covariate in this model, it may just be a mistake in the wording? From the results I gathered that you tested the effect of Ta on Tb (i.e. their correlation), but this sentence here reads as if you tested the effect of Tb on Ta.

Line 278: Unclear to me what function you used, mgcv is a package, not a function, as far as I am aware. Did you use a “gam.check” or something similar to test for the number of knots in the smooth? For all of the non-linear relationships tested, it would be good if you describe the number of knots used, and the type of spline fitted. This will affect the model interpretations.

Line 279: While I believe polynomial is correct, it’s common to describe the specific type of spline used in a GAMM. The standard smooth function s() is a thin-plate regression spline with either automatically set knots (k) or predefined knots. For time of day it is common to use a cubic circular or cubic regression spline, or a cosinor function. How were the different splines fitted? Please describe this in more detail rather than just “polynomial”.

Line 286: here, also please be clear that you considered the best fit model to be the one with the lowest AICc score. It should be obvious, but important to be explicit about your selection criteria.

Line 287: I am glad to see this part of the methods described

Results

Line 323: here you define the AICc selection criteria, but you did not explicitly state that the model with the lowest AICc score was considered the best fit (a small, but important detail)

Line 342-344: Suggest this sentence to something simpler, e.g. “There was a significant effect of […] after controlling for temporal autocorrelation.”

Line 246: Fig 2 still does not show the relationship between Ta on Tb (I see you have altered in the Figure legend), but how they fluctuate together over time. Do you mean to refer to figure 5? If so the order of the figures should be rearranged to match the order in which they are referred to in the results.

Lines 346 and 357: Here, you describe the best fit models with only marginal delta AICc values, 0.58 and 1.38 (a common criteria is that a model is considerably better if dAICc < 2 or 5). Since you did not predefine the criteria for selecting the best model (i.e. your dAICc threshold is not explicitly stated in the methods), it’s not automatically clear why GAMM2 is considered significantly better than GAMM3. Perhaps you can soften the wording to “a marginally better fit” rather than “best fit”? Since I can’t see the model results (tables 1 and 2 are missing), and figure 2 does not show the spline/regression results, it’s not possible to evaluate this part of the results.

Line 364: I believe you mean to refer to figure 3 here? It seems the figures may need to be rearranged

Line 366: great that you provide the summarised bootstrap results from RMR projections. Is there a reason why you did not do the same for the Tb results?

Discussion

Line 383: A bit unclear what you mean here, did you only have 3% overlap in Tb records with those recorded for wild sloths?

Line 471: Of course, metabolic rates cannot escalate indefinitely, so this sentence reads a little awkwardly. Perhaps this can be rephrased so as to highlight the impact of the response in high-altitude sloths? Something along the lines of “... are likely to be more constrained in their ability to adapt to a continuously warming climate”? (only a suggestion)

References:
You have several duplicated (and triplicated) references which needs amending:
- Colwell et al. 2008
- Deutsch et al. 2015
- Foley et al. 1995
- IPCC 2007 (also, you may consider using the latest IPCC report)
- Norin and Metcalfe 2019
- Parmesan 2006
- Root et al. 2003
- Seebacher et al. 2014


Figures
Figure 3: I would have preferred to see the 95% confidence intervals plotted together with the predictions, if possible. This would emphasize the uncertainty around them…

Figure 5: The first two lines here are just a description of the results. Please describe the figure instead, i.e., “The relationship between Ta and Tb in high- and low- altitude sloths…” etc

Supplementary material

Supplementary figures: please provide captions for the supplementary figures.

Table S2: Please describe any non-SI abbreviations (RMR, Ta, Tb) in your table in the caption so the table is self-standing. Also, perhaps you could include “measured RMR” and “predicted RMR” in the table to better distinguish between the allometric predictions and measured values?

Table S3: I suggest that you change “bootstrap analyses” to “bootstrap approach” as in your main text. Also, please include CI in parentheses in the table caption, i.e. “The confidence intervals (CI) were calculated […]”

A small detail about the table references: ideally, the tables should appear in the order they are referenced, so technically Table S2 before Table S3. Since table S3 show results and are not methods, perhaps you can remove the reference in the methods (line 299), so that Table S2 is referenced prior to Table S3 in the main text? I am aware that this is a small detail, but it improves the readability as one is not left thinking something is missing…
Figure S3: the difference between the two groups would be more striking if you use the same scale on the x-axis, or potentially in the same plot, so that it’s easier to see that the projected Tb for high-altitude sloths is actually higher than that for low-altitude sloths

---

## Round 0.3 · accepted · Accept

Dear Dr Cliffe and colleagues,

Many thanks for submitting the revised manuscript so promptly and for addressing reviewers' comments. After reviewing your revision, I am happy to accept the manuscript in its current form. Thank you again and congratulations!